# Variability of Human rDNA

**DOI:** 10.3390/cells10020196

**Published:** 2021-01-20

**Authors:** Evgeny Smirnov, Nikola Chmúrčiaková, František Liška, Pavla Bažantová, Dušan Cmarko

**Affiliations:** Institute of Biology and Medical Genetics, First Faculty of Medicine, Charles University and General University Hospital in Prague, 128 00 Prague, Czech Republic; nikola.chmurciakova@lf1.cuni.cz (N.C.); frantisek.liska@lf1.cuni.cz (F.L.); pavla.bazantova@lf1.cuni.cz (P.B.); dusan.cmarko@lf1.cuni.cz (D.C.)

**Keywords:** human rDNA, sequence variability, mutations, copy number

## Abstract

In human cells, ribosomal DNA (rDNA) is arranged in ten clusters of multiple tandem repeats. Each repeat is usually described as consisting of two parts: the 13 kb long ribosomal part, containing three genes coding for 18S, 5.8S and 28S RNAs of the ribosomal particles, and the 30 kb long intergenic spacer (IGS). However, this standard scheme is, amazingly, often altered as a result of the peculiar instability of the locus, so that the sequence of each repeat and the number of the repeats in each cluster are highly variable. In the present review, we discuss the causes and types of human rDNA instability, the methods of its detection, its distribution within the locus, the ways in which it is prevented or reversed, and its biological significance. The data of the literature suggest that the variability of the rDNA is not only a potential cause of pathology, but also an important, though still poorly understood, aspect of the normal cell physiology.

## 1. Introduction

In human cells, ribosomal DNA (rDNA) is arranged in ten clusters of multiple tandem repeats. These clusters are known as Nucleolus Organizing Regions (NORs) and situated on the short arms of the acrocentric chromosomes (#13, #14, #15, #21, #22). According to standard description, each rDNA repeat typically consists of the 13 kb long ribosomal part and the 30 kb long intergenic spacer (IGS). The ribosomal part transcribed by RNA polymerase I (pol I) produces a single large precursor, which includes three genes, separated by internal transcribed spacers, ITS1 and ITS2, and flanked by external spacers, 5′ ETS and 3′ ETS (Figure 1A,B) [1,2,3,4,5,6,7,8,9,10]. The transcription levels of these genes are very high throughout interphase. The primary transcript undergoes processing into 18S, 5.8S, and 28S rRNAs of the ribosomal particles. IGS is a large and complex locus with multiple repetitive elements. The largest of them are 2 kb long LR1 and LR2 repeats, situated at the middle of IGS and each containing two *Alu* sequences. There are also R-repeats (blocks of about twenty 500–800 bp units) positioned downstream of the 3′ETS and containing abridged termination signals; butterfly repeats (units of approximately 300 bp); 90 bp repeats (blocks of simple motifs CTTG and CTTT separated by CGTG); *ActcA* repeats (blocks of three tandemly arranged 84 bp units); non-tandem repeats *abc* (58 bp long) and *bc* (52 bp long); microsatellites (2–6 bp long, repeating at least three times); and transposable elements, among which *Alu* are predominant [2,11,12].

The best studied part of IGS is the ribosomal gene promoter, whose key elements are the upstream control element (UCE) and core promoter elements (CPE) [13]. Upstream of the promoter lies the enhancer, whose precise location in human cells is not established [14,15]. Recent studies suggest that IGS has a complex functional organization. RNA-seq analysis of human and murine rDNA has revealed a specific pattern of low-abundance expression over the entire spacer region [15,16,17,18,19,20,21]. One prominent transcription unit includes a 2 kb long segment at the 3**′** end of IGS, and probably stretches into 5**′** ETS area [15,20,22,23,24]. This region is transcribed by pol I and is involved in rDNA silencing [10,22,25,26,27,28,29,30,31,32]. A still-larger regulatory locus covers part of the ribosomal gene area, promoter, and enhancer. It is transcribed by pol II in the anti-sense direction, and its 10 kb product called “promoter and pre-rRNA anti-sense” (PAPAS) causes the repression of rRNA synthesis through different pathways [33,34,35,36,37,38]. A number of IGS sequences, some of which have already been identified, participate in the “nucleolar detention” [39], which takes place under stress conditions: the non-coding RNAs (ncRNAs) produced by these sequences recognize, detain within the nucleoli, and temporarily inactivate certain proteins equipped with a “detention signal” [18,39,40,41,42]. *Alu* sequences scattered over the spacer also produce transcripts, which may be involved in the maintenance of the nucleolar structure [43,44]. Other regions of IGS have been recently described as potentially functional [20,45].

Thus, rDNA is characterized by the multiplicity of copies and complex structure of each copy, which includes many other repetitive elements. But, a no less important feature of this genomic locus is its peculiar instability, producing a great number of variants [35,46]. The differences in the rDNA sequence have been found among different units belonging to one cluster, or NOR [47,48], among different NORs [48,49], among different cells and tissues [44,49,50,51], and among different human individuals and populations [51,52,53]. The multiplicity of the copies and their variability hamper the study of rDNA so much that this locus has been excluded from the human genome project [54,55]. The existing databases present only averaged assemblies of the locus processed by special software [15,56,57,58].

## 2. Detection of the rDNA Variations

Efficient comparison of different species has been made possible by the assembly of rDNA repeats from whole-genome data, but this procedure fails to detect major variations within species and individuals. Cloning methods, such as transformation-associated recombination (TAR), which provide a few copies of rDNA for sequencing, partly solves the problem, yet each of these copies still contains internal repeats that make assembly difficult [56]. Recent advances in sequencing technology have helped to obtain reads longer than 100 kb, so that entire rDNA units could be captured in one read [59]. Although the error rate of long-read sequencing is relatively high, single-molecule technologies such as PacBio achieve a very high consensus precision, for the errors they produce are mostly random, and can be corrected if sequencing depth is sufficient [60]. A combination of cloning with multiple sequencing methods allowed characterization of rDNA variability in a single chromosome of one individual [61].

However, sequencing methods encounter serious difficulties in dealing with large scale variants of rDNA locus. Thus, the identification of palindromes based on the reads is uncertain [61]. A kind of alternative to the sequencing is provided by fluorescent in situ hybridization on the spread DNA fibers (fiber-FISH). This method allows to visualize abnormal rDNA repeats, to measure the length of regions revealed by FISH probes, and to study distribution of methylated and other modified bases in the locus [47,62,63,64,65].

Another method is based on extraction of the rDNA clusters (NORs) by digesting the rest of the genome with a set of restriction enzymes [53]. The intact clusters obtained from the genomes are separated by size in a pulsed-field gel, transferred to blotting membrane, and hybridized with an rDNA-specific radiolabeled probe. The distribution of the fragments shows distinct reproducible patterns called electrophoretic fingerprints. This method, despite its low resolution and dependence on such contingencies as restriction site deletion, may be used for the study of individual large-scale variability of rDNA in normal human patients and in pathology.

A particular problem is the determination of the number of rDNA repeats per cell or chromosome, for the repeats usually differ significantly in length and structure. Pulsed-field gel electrophoresis and Southern blotting may be used for this purpose, but recently qPCR-based procedures have been preferred [66]. Fluorescent probes are employed to monitor the progress of PCR amplification of a given target at each cycle. The droplet digital PCR (ddPCR) has been increasingly used to measure rDNA copy number. In this method, the diluted DNA samples are loaded into a plate of wells, each containing 1–2 molecules. The concentrations of the targets of interest are then determined by counting the number of fluorescently positive and negative droplets in the sample [67,68]. Since regions rich in either GC or AT bases are underrepresented in the chain reaction assays, the number of rDNA copies, e.g., in a single individual, has been estimated by special statistical methods which take the sample-specific GC content bias into consideration [51]. However, it should be noted that the models developed in these works are based on certain premises. For instance, a Poisson distribution of the GC content along the locus is postulated. Although this seems plausible, the premise should be well tested, for it is a strong statement, implying a high randomness of the rDNA structure.

Microscopy also may be helpful in the study of rDNA copy number. Comparative data about distribution of this number among individual acrocentric chromosomes have been obtained by FISH with rDNA probes on the preparations of spread mitotic chromosomes, for the intensity of fluorescent signal belonging to each NOR correlates with the number of repeats [49,69]. With proper calibration of the signal, this method may also be used for precise repeat counting, by analogy with the quantitative analysis of nascent transcripts by single-molecule FISH [70]. Additionally, silver staining of the spread mitotic chromosomes may be used for the quantification of the transcriptionally active genes [49,71,72,73].

## 3. Causes and Kinds of rDNA Variability

Like the rest of the genome, rDNA is subject to base substitution, slipped strand mispairing (SSM), and non-SSM mutations. The observed variability, resulting from replication errors or recombination, is most frequently represented by single-nucleotide variants (SNV), short insertions and deletions (indels), as well as variable numbers of the short, repeated elements [74,75]. The potential breakpoints, where DNA sequence tends to undergo these and other changes, are often represented by microsatellites, (TC)n, (TG)n, and short (3–5 bp) poly-N clusters [14,76,77]. Approximately 7.5 variants per kb have been found in rDNA, which is comparable to typical estimates of variation across the genome [61].

However, factors peculiar to rDNA modify the character of its variability. The multiplicity of the repeats, which include the 43 kb unit itself as well as its components, lays the foundation for the mutability of the dosage or copy number [78]. On average, human rDNA have between 200 and 600 copies, but the number may vary by nearly two orders of magnitude [51]. In one study, the number of the ribosomal gene copies per cell varied in different individuals: from 67 to 412 for 18S ($\bar{x}$ = 217 copies), from 9 to 421 for 5.8S ($\bar{x}$ = 164 copies) and from 26 to 282 for 28S ($\bar{x}$ = 118 copies) [52]. Here, although the pair-wise correlations, e.g., between the numbers of 28S and 18S, were high, the discrepancies of both mean values and extremes, if they did not proceed from an error of measurement, indicate that a significant portion of the rDNA units may lack 28S, 5.8S or 18S components [52]. But, if all these incomplete units are transcribed and processed, it is not clear how the stoichiometry of the ribosomal components is maintained. It also seems extraordinary that a person has only nine ribosomal genes per cell; this number would be normal for prokaryotes [79], but homoiothermal animals usually have about 200 copies per genome [5].

The abundance of repeats in the rDNA cluster increases the probability of recombination [80]. Besides, rDNA contains multiple extensive regions of [CT]n, [CTTT]n, [TG]n, as well as G- and GC-rich segments, which mark the hotspots of recombination [50,53,81,82,83,84,85,86]. The recombination events can occur between adjacent (within one 43 kb unit) or remote (across the units) repeats. In particular, the shell of chromatin surrounding nucleoli (perinucleolar chromatin) contains many repetitive sequences, such as LINE/L1 and SINE/*Alu*, which are regarded as candidates for interaction with rDNA [87,88,89,90]. It was supposed that a recombination between the perinucleolar chromosomal regions and the closely situated IGS loci producing ncRNAs may affect the organization of the nucleolar DNA [14]. Generally, recombination events lead to deletions, duplications or inversions, e.g., inversions producing head-to-head orientation and sometimes resulting in elimination of the coding sequences [61]. Most frequently, the unequal chromosomal exchange causes copy number reduction, although an increase in this number is also possible [91,92]. The loss of repeats after breaks in the rDNA mostly occurs through recombination between sister chromatids or rDNA repeats on different chromosomes [93]. The locus may therefore be restructured very quickly. The rate of human meiotic chromosome exchanges in rDNA was estimated as 11% per generation per gene cluster (95% confidence interval: 6.4–20%), which, alone, perhaps could account for the observed differences among the human individuals [93]. There seems to be an intricate dynamic relation between replication errors, recombination hotspots, and repeats. Apparently, in each S phase, the replication fork progress is sometimes arrested by secondary DNA structures, which is especially abundant in the regions of repeats [74,75]. The fork stalling often leads to the formation of double-strain breaks (DSB), which can then be repaired by intra- or inter-chromosomal recombination events with an ensuing change in copy number, e.g., the number of coding sequences [35,52,94].

The same features, namely, frequent recombination, inconstant copy number, and potentially high level of replication errors, are shared by the other repetitive loci of human genome, telomeres and centromeres [80]. However, rDNA is distinguished from these regions by its high intensity of transcription throughout the interphase, which increases the risk of instability caused by the collision of the replication and transcription machineries [7,74,75]. Moreover, transcription itself is believed to be mutagenic, because it is accompanied by a torsional stress, and the topoisomerases relieving this stress create nicks and DSBs in the template DNA [95]. It has been suggested that the dosage variation induced by the intensive transcription may be responsible for the increased mutability of the rapidly proliferating cancer cells [78].

In addition to the variable copy number, other large-scale variations involving regions of 2 kb and longer seem to occur in human rDNA on a scale not previously envisaged (Figure 2). Thus, an insertion of about 2 kb length could be regularly observed in IGS between positions 22,703 and 22,714. Large palindromes were also found in the sequencing reads [61]. However, the large-scale variability of rDNA is most clearly revealed by fiber-FISH staining with two DNA probes covering adjacent regions of the ribosomal part and IGS, respectively [47,62,63,64,65]. The above-cited studies indicate that about one third of the rDNA units exist in the inverted or palindromic orientation in both tumor-derived and primary cell lines, as well as in normal adult and fetal cells [47,61]. The unorthodox arrangement may result from crossover at *Alu* sequences and the production of single-stranded loops with an extension of the opposite strand. Since various kinds of palindromes often appear as clusters of similar structures, their formation probably requires DNA amplification steps [47]. Microscopic measurements on the fiber-FISH preparations showed an extraordinary variability in the length of IGS and ribosomal parts of rDNA. Thus, although the normal length of human rDNA spacer is believed to be about 30 kb, data obtained on several hundred units indicated an average IGS length of 34.2 ± 5.4 kb, with individual IGS units ranging from 9 to 72 kb [47]. Moreover, the ribosomal area had approximately normal length in only 30–40% of units [61]. These data suggest that the large-scale arrangement of human rDNA typically deviates from what used to be regarded as the norm.

The presence of multiple transposable elements is another feature of rDNA closely connected to its variability. These elements have been called the greatest source of repetitive DNA in the eukaryotic kingdom [35]. Human rDNA is particularly rich in the *Alu* belonging to retrotransposons, which move in a copy and paste manner by way of an RNA intermediate [35,96]. Together with the transposable element pseudo-cdc27, *Alu* constitutes 26% of the IGS length [11]. *Alu* repeats often contain simple short (3–5 bp) repeats and their structure is very mutable. Moreover, the most active break points in rDNA are usually grouped upstream and downstream of *Alu*, although not all *Alu* are predisposed to breaks [14]. Until recently, the transposons have been referred to as “genetic parasites” or “selfish” DNA, which can produce various genetic alterations and disrupt normal gene expression as a consequence of their transposition [97]. However, closer scrutiny suggests the need to revise this view. Although the length and copy numbers of *Alu* elements vary, most of them are incomplete and unable to migrate. Thus, they are typically found in the same positions as, e.g., regular expressed sequence tags (EST) in the LR1 and LR2 regions of IGS [11]. Recent studies indicate that *Alu* participates in the structural organization of nucleoli. The depletion of aluRNAs disrupts nucleolar structure and impairs rRNA production, while the overexpression of aluRNAs increases both nucleolus size and rRNA production. Microinjection of these RNAs into α-amanitin-treated cells-initiated re-assembly of the previously disrupted nucleoli. Moreover, after the knockdown of aluRNAs, nucleolin and pol I were dispersed [43]. aluRNA was also found to associate with nucleolin and nucleophosmin, forming RNP complexes which may play a key role in the post-mitotic assembly of NORs [98].

The abundance of non-canonical DNA structures, especially R-loops (the complexes of a DNA:RNA hybrid and the associated non-template single-stranded DNA), also has a significant impact on the variability of the rDNA locus. A new sequencing technique, called single-molecule R-loop footprinting (SMRF-seq) coupled with PacBio sequencing, allowed to map pol-I-derived R-loops at the single-molecule level and showed that they are non-randomly distributed in the human rDNA. Most frequently, R-loops span from 100 to 450 bp, but at the 18S gene they can reach 2 kb in length [99]. Usually, these structures are regarded as both the causes and consequences of DNA instability, but a recent study suggested that they may be involved in regulation of the copy number in the NOR [100]. Accordingly, some authors propose distinguishing between “scheduled regulatory” and “unscheduled harmful” R-loops [101]. The data of a recent study indicate that antisense transcription by nucleolar RNA polymerase II generates an “R-loop shield” at IGS, which protects nucleolar organization by suppression of disruptive ncRNAs transcribed by pol I [102].

## 4. Distribution of the Variability within the rDNA Unit

As a whole, IGS is more variable than the ribosomal part of the locus (Figure 1C). The structure of the spacer often varies in different rDNA repeats, even within the same NOR [2,3,11,47,50,53,95,103,104,105,106]. About 70% of the total number of small variants (SNV, insertions, and deletions) belong to IGS. SNVs and deletions are particularly numerous there, but insertions, which are relatively sparse in the rDNA, appear less frequent in the spacer than in the ribosomal part [11,61]. The putative DNA breaks are often concentrated in discrete areas of IGS with the length of 500–900 kb, surrounded by analogous repetitive elements (microsatellites, simple sequences, and whole or abridged *Alu* elements) [14]. The breakpoints seem to be especially abundant at regions of large repeats of LR1 and LR2 [61]. The rDNA promoter region also shows some variability, including SNVs, short inversions, and deletions, but they are rare in the core promoter region [61]. Variants near the transcription start site of the pre-rRNA are supposed to have an effect on the expression [107,108].

The coding rDNA regions, usually described as conservative and identical, proved to be variable as well [11,48,50,51,61,109,110]. In the GC-rich region of 28S rDNA, especially towards its 3**′** end, around the restriction site of HincII enzyme, the variability is comparable to that of IGS, and the HincII site itself is often absent [48,51,61,109,111]. Generally, among the variants of the coding regions, the single-nucleotide substitutions appear most frequently; there are also supplementary microsatellite clusters, and, more seldom, extended deletions [14,76,112]. Extraordinary numerous DNA:RNA hybrids, which may be a source of genomic instability, were discovered in 28S and 18S rDNA [113]. The variations in the human rRNA sequence involve no less than 20% of the nucleotide [51]. Most variants of the ribosome do not alter its structure and appear in expansion segments, which are known as sequence-variable regions in other species. Only a few are predicted to alter RNA structure significantly [61]. On the other hand, it was supposed that rRNA sequence variants in ribosomal-protein-binding domains may be connected to the translation of distinct mRNAs, thus affecting the whole pattern of gene expression in the cell [61,114].

The transcribed spacer (TS) regions of rDNA also have to preserve their stability, since they contain the cleavage sites [2]. However, variants, mostly short deletions, appear in all parts of the TS, and the mean percentage of these variants proved to be roughly the same as in IGS [61]. Moreover, some of them lie near processing sites, e.g., sites 01 and 1 in 5′ETS, and between sites 3 and E in ITS1, and might influence pre-rRNA processing through modification of RNA folding [61,110].

## 5. Stabilizing rDNA Locus

Despite its potentially high instability and variability, human rDNA is able to maintain its architectonics and function in the course of multiple cell generations. Apparently, many sites of DNA damage may be repaired, as in other parts of the genome. However, it is believed that the same property of rDNA that makes the locus unstable, namely the great number of copies, may also help to stabilize the locus through unequal crossing-over and gene conversion [14,50,115,116]. The homology-dependent DNA repair can take place during meiosis, as well as in S/G2 and (less often) in G1 cells [117]. Copies in both the cis and trans positions are used for the repair [115]. Thus, the most dangerous rDNA lesions, such as double-strand breaks caused by radiation, may be repaired by homologous recombination with another unit of a neighboring repeat. In this process, the intervening units, which form a loop between the donor and acceptor site, may be excised, reducing the total number of rDNA repeats [46]. It has been believed that the efficiency of this restoration is enhanced by the presence of numerous transcriptionally silent, heavily methylated, and supposedly stable units in each transcriptionally active cluster [1,6,10,15,28,118,119,120,121,122,123,124]. On the other hand, the transcriptionally silent repeats may serve to prevent unequal homologous exchanges, causing instability [1,6,10,88,121]. In yeasts, the length of the rDNA cluster expands and contracts in a cyclic manner. Transcription from the additional promoter E-pro enhances recombination events by removing cohesin complexes, which may change the number of copies, and it seems likely that the human spacer promoter plays a similar role [46]. However, this does not explain how the desired outcome of recombination is ensured, and especially how the normal repeats are selected; the task seems particularly difficult since the units of the abnormal large-scale structure are abundant, if not predominant, and often appear in tandem [47]. At any rate, purely random recombination could hardly prevent accumulation of errors. Moreover, the amplification of rDNA arrays, which is not necessarily abnormal, may hinder faithful genome replication and repair [78].

Perhaps there is a dynamic balance in the cell between the production and elimination of abnormal rDNA units. For instance, a steady-state level of palindromes may be maintained based on a balance between their generation, expansion and elimination [47]. It seems that the formation and suppression of the R-loops is also subtle regulation. In a similar manner, it seems that multiple proteins and complexes, including RNA–DNA helicases, RNase H1/2, and RNA-processing factors, participate in a subtle control of the R-loops formation and suppression [100,102,125,126].

In certain conditions, it may be beneficial for the cell to reduce the number of rDNA copies, for this would diminish replication stress, making it easier for cells to progress through the S phase of the cell cycle [78,127]. For instance, a loss of rDNA repeats in human cancer genomes may be adaptive [78]. The copy number also decreases with age [128,129,130]. Remarkably, the cultured human fibroblasts in the course of replicative senescence lost hypermethylated copies of rDNA, which had been previously regarded as a potential remedy for instability [131]. On the other hand, it was suggested that such “cleansing” of the locus induces senescence [46].

The structure of nucleoli restricts interaction of the rDNA repeats and affects the stability of the locus. During the interphase, a section of one repeat, which includes three coding sequences, together with their promoter and terminator, probably occupies one nucleolar compartment, known as the Fibrillar Centre/Dense Fibrillar Component (FC/DFC) unit, whereas the rest of the IGS regions are accommodated in the surrounding Granular Component (GC), composed chiefly of the ribonucleoprotein particles. Thus, each transcription unit is isolated from the others; perhaps only the sister units produced in the course of replication may be engaged in the crossover events during interphase [8]. Besides, collision of the DNA polymerase and RNA polymerase machineries in the S phase seems to be prevented by suspension of the transcription in those FC/DFC units where DNA is currently replicated [7].

## 6. Significance of the rDNA Variability

The role of rDNA variability in human pathology remains unclear. The abundance of the apparently abnormal units suggests that some of them, provided that they preserve an intact promoter and at least one coding region, can produce functional RNA components of ribosome. On the other hand, it has been supposed that persistent rDNA damage signaling and structural rearrangements after erroneous repair, followed by the disorder of transcription and ribosome dysfunction, may trigger disease. Many studies show a correlation between human disorders, especially cancer, and rDNA variability. Thus, in epithelial cancer cells, a significant increase in SNV was found in the region between −388 and +306 kb (including part of the promoter and 5**′** ETS regions) [110]. Predisposition to cancer, premature aging, and neurological impairment in ataxia-telangiectasia and Bloom syndrome, coincided with increased cellular rDNA repeat instability [53,110,132,133,134]. Considerable expansion or contraction of the rDNA locus are usual symptoms of malignant transformation [53,78,133]. Micro-RNAs produced by certain IGS regions may serve in cancer diagnosis [135]. Nevertheless, it has not been proved that rDNA instability can cause severe health problems [127,136]. This instability seems to be a consequence rather than a cause of the pathology. In fact, the genesis of most known ribosomopathies is attributed to mutations in ribosomal protein genes, not to abnormalities of rRNA [127,137].

Most of the rDNA variants are probably low-impact mutations. Application of the method known as electrophoretic karyotype or electrophoretic fingerprints (see Section 2) to the genomes of normal human patients revealed distribution of the rDNA clusters peculiar to each human individual by length. On the gels, there were eight to ten bands per person, and the clusters of rDNA ranged in length from ∼70 kb (less than two standard repeats) to >6 Mb (more than two medium-sized NORs) [53]. These experiments, among other things, indicate that the existence of multiple variants may be regarded as a normal feature, reflecting its complexity [47].

At least one kind of rDNA mutability, namely, the loss of repeats, has been regarded as adaptive. It apparently facilitates genome replication and accelerates proliferation [78]. A reduction in the repeat number artificially induced by zinc-finger nuclease genome-editing increased the proportion of the transcriptionally active ribosomal genes [138]. According to another hypothesis, the reduction in the repeat number with age induces senescence in order to protect cells against genome instability and cancer [46]. On the other hand, it has been found that rDNA dosage is correlated with the expression of several proteins that modify chromatin. There was also a correlation between the number of the rDNA repeats, the expression of mitochondrial-related genes, and the abundance of mitochondrial ribosome genes themselves [52]. These data suggest that the observed instability of rDNA locus may be associated with diverse functions, which still remain unexplored.

A part of the NORs, or individual ribosomal genes belonging to one cell, are maintained in a transcriptionally silent state [1,6,10,49,118,121,122,123]. However, although it seems obvious that rRNAs cannot be transcribed from severely disordered units, the impact of the sequence upon the transcription question is still unknown. As we have seen, the variability of the locus is multiform, and it remains unknown whether certain DNA sequences cause the differentiation of the genes into active and silent, that is to say, why some of the rDNA repeats acquire a chromatin structure that is favorable for their transcription, while the rest remain in the silent state. On the other hand, the observed abundance of the large-scale variation revealed by fiber-FISH technique suggests that the apparently abnormal repeats could also produce functional RNA components of ribosome.

## 7. Conclusions

Human rDNA is often described as a particularly instable locus with numerous hot spots of recombination. Both IGS and the intensively transcribed ribosomal regions are involved in various kinds of mutability which spectacularly alter the normal, or supposedly normal, structure of the locus. However, the biological significance of this prominent phenomenon remains unclear; even its role in human pathology has not been established. Some danger may perhaps proceed from the abundance of small mutations, which are for the most part low-impact negative or neutral, and can accumulate, especially within the IGS region, and substantially contribute to the mutational load of the human genome [139,140]. However, on the other hand, the extraordinary variability observed in the normal human genomes shows that most cells can manage it. A dynamic balance between the generation and elimination of variants, especially the number of repeated elements, prevents dramatic alteration of the locus [46,141]. Moreover, changes in the rDNA structure seem to be involved in many vital processes, including gene expression and the function of mitochondria. All this suggests that the variability of the rDNA locus is an important, though still poorly understood, aspect of the normal cell metabolism.

## Figures and Tables

**Figure 1 cells-10-00196-f001:**
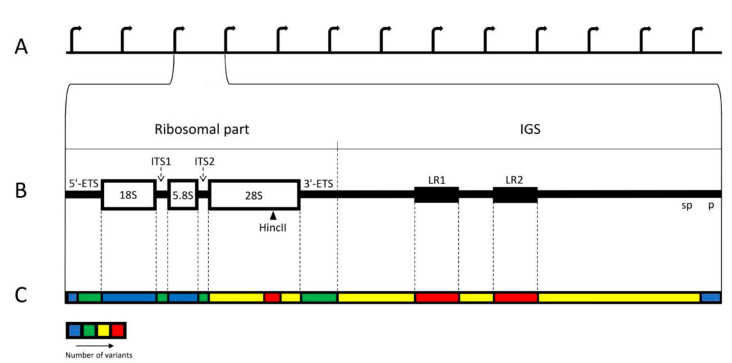
Distribution of the variability among the regions of human rDNA unit. (**A**) Standard structure of the rDNA cluster. (**B**) Individual rDNA unit. The unit consists of the ~13 kb long ribosomal part and ~30 kb long IGS. The ribosomal part includes three genes: 18S, 5.8S, and 28S, coding for the respective rRNAs; p—promoter; sp—spacer promoter; two external spacers: 5′ETS and 3′ETS; two internal spacers: ITS1 and ITS2. Restriction site HincII in the 28S region is marked by arrowhead. LR1 and LR2 represent two large repeats, and each of them contains two *Alu* sequences. (**C**) Thermogram showing relative abundance of variants and breakpoints in various regions of the unit. The least variable are two genes and a region including core and upstream control elements of the promoter. The most variable are both LRs and a part of 28S gene containing HincII site.

**Figure 2 cells-10-00196-f002:**
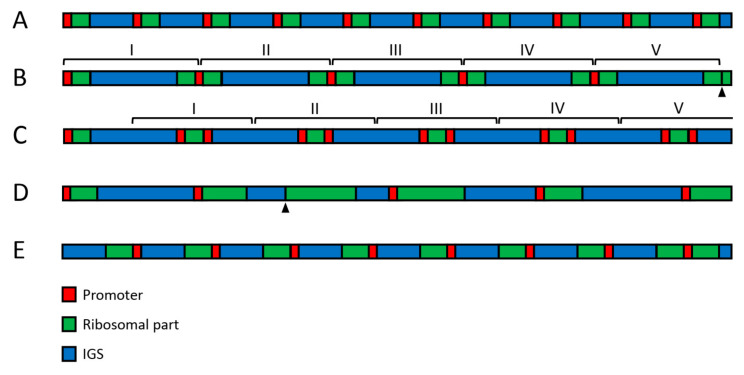
Large-scale variability of human rDNA. (**A**) The standard structure, in which promoter (red) is followed by ~13 kb long ribosomal region (green) and ~30 kb long IGS (blue). This pattern does not seem to be very frequent, but it is preserved in the cell generations. (**B**) Five 3′-3′ palindromes arranged in a cluster. (**C**) one normal unit followed by five 5′-5′ palindromes arranged in a cluster. (**D**) Units with variable length of ribosomal part and IGS. (**E**) Inversion involving six units. The last unit at the right has standard orientation. Arrowheads indicate deleted promoters.

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
