# Peer review of "Variability of Human rDNA"

_cells, 2021, doi:10.3390/cells10020196_

Round 1

Reviewer 1 Report

This is an excellent review of the current knowledge around ribosomal RNA structure and variability. It includes references from old and new papers and will be a valuable resource to anyone looking to understand this field. It is also very clear, logically structural and easy to read. I would strongly endorse the publication of this manuscript. Here is a short list of very minor suggestions and comments:

Line 55: "starts at about -2kb and stretches probably beyond the 3' end of IGS" This is a little unclear. -2kb from where? And in which direction? I can make a good guess what the authors mean, but some further clarity would be appreciated.

Lines 134-137: If I understand this correctly, are the authors suggesting that individual (28S, 18S, 5.8S) rRNAs are being transcribed from incomplete 45S pre-rRNA? Or just that some these sequences are a different frequencies in the DNA while not being transcribed? If it is the former, that seems to be a very novel idea, and not one suggested by the cited Lemos paper, so the authors should make it clear that this is their suggestion. The way that it is cited implies that this is the original authors' viewpoint. However, if it's the latter, perhaps the sentence could be clarified by using "ribosomal units" instead of "transcription units" since they use that term in Figure 1.

Given the emergence of drugs targeting Pol I, the authors might want to note that rDNA repeat deletion can lead to an increase in the active:silent repeat ratio and increased sensitivity to Pol I inhibition (Son et al., Front Cell Dev Biol, 2020).

Author Response

we thank you for your review report of our manuscript Variability of human rDNA and your important comments and provided references. We discussed them all and revised as follows (the remarks requiring reply are in bold font):

Line 55: "starts at about -2kb and stretches probably beyond the 3' end of IGS" This is a little unclear. -2kb from where? And in which direction? I can make a good guess what the authors mean, but some further clarity would be appreciated.

In the revised version of our manuscript we edited the indicated passage thus: “One prominent transcription unit includes a 2kb long segment at the 3’end of IGS and stretches probably into 5’ ETS area“

Lines 134-137: If I understand this correctly, are the authors suggesting that individual (28S, 18S, 5.8S) rRNAs are being transcribed from incomplete 45S pre-rRNA? Or just that some these sequences are a different frequencies in the DNA while not being transcribed? If it is the former, that seems to be a very novel idea, and not one suggested by the cited Lemos paper, so the authors should make it clear that this is their suggestion. The way that it is cited implies that this is the original authors' viewpoint. However, if it's the latter, perhaps the sentence could be clarified by using "ribosomal units" instead of "transcription units" since they use that term in Figure 1.

Indeed, the data referred to by the reviewer concern the DNA sequence, without relation to the transcription. In the revised version, to avoid ambiguity, we change “transcription units“ to “rDNA units.“ At the end of the paragraph we added another commentary: “But, if all these incomplete units are transcribed and processed, it is not clear how the stoichiometry of the ribosomal components is maintained. Also, it seems extraordinary that a person has only 9 ribosomal genes per cell; this number would be normal for prokaryotes (Klappenbach et al, 2000, PMID: 10742207), but the homoiothermal animals usually have about 200 copies per genome (Long and Dawid 1980, PMID: 6996571).”

Given the emergence of drugs targeting Pol I, the authors might want to note that rDNA repeat deletion can lead to an increase in the active:silent repeat ratio and increased sensitivity to Pol I inhibition (Son et al., Front Cell Dev Biol, 2020).

This seems to be an important finding. We place it in the section 6, paragraph 3, of the revised version: “Reduction of the repeat number artificially induced by zinc-finger nuclease genome-editing increased the proportion of the transcriptionally active ribosomal genes (Son et al, 2020, PMID: 32719798).“

We hope that you find these revisions sufficient.

Reviewer 2 Report

Overall, I feel that this is a very nice review on variability of human ribosomal DNA. As far as I can ascertain most of the relevant studies are referenced, the review is informative and well written. 

As a minor suggestion I recommend considering discussing two additional R-loops related studies in the text in case the authors find it appropriate: PMID: 32669707; PMID: 25073155

Author Response

we thank you for your review report of our manuscript Variability of human rDNA and your important comments and provided references. We discussed them and revised as follows (the remarks requiring reply are in bold font):

As a minor suggestion I recommend considering discussing two additional R-loops related studies in the text in case the authors find it appropriate: PMID: 32669707; PMID: 25073155

Suggested articles are cited in the sections 3 of the revised version of our manuscript:

”Accordingly, some authors propose to distinguish between “scheduled regulatory” and “unscheduled harmful” R-loops (Niehrs and Luke, 2020, PMID: 32005969). Data of a recent study indicate that antisense transcription by nucleolar RNA polymerase II generates an “R-loop shield” at IGS, which protects nucleolar organization by suppression of disruptive ncRNAs transcribed by pol I (Abraham et al, 2020, PMID: 32669707).“

And in the section 5:

“In a similar manner, formation and suppression of the R-loops seem to be regulated. Multiple proteins and complexes, including RNA–DNA helicases, RNase H1/2, and RNA processing factors, participate in a subtle control of the R-loops. formation and suppression (Matson and Zou, 2020, PMID: 32611612; Salvi et al, 2014, PMID: 25073155; Abraham et al, 2020, PMID: 32669707; Vydzhak et al, 2020, PMID: 32446803).”

We hope that you find these revisions sufficient.

Reviewer 3 Report

In the manuscript, the authors have reviewed the variability of ribosomal DNA, the region of human genome that attracts growing attention. The variability is regarded in two aspects, which can be combined with each other: non-standard structure of the ribosomal repeats such as SNP, indels, etc., and variability in copy numbers of the (identical) repeats.

The review is logically structured, and cites interesting reports I could learn some novel information from. I found no errors in the text, except for one doubtful reference (see below), and  the only thing remaining to do is adding some references the review would be incomplete without. Therefore, I recommend to accept the manuscript after two minor revisions.

Firstly, there is a questionable reference for the data related to the total copy number of ribosomal repeats in human genome. This is reference #52 with copy number data in lines 133-134: 'In one study the number of the ribosomal gene copies per cell varied in different individuals… from 9 to 421 for 5.8S… and from 26 to 282 for 28S.' Below, in lines 135-136, the authors expressed slight scepticism that the discrepancies could '…proceed from an error of measurement', but the data was generally presented as feasible. My opinion is that these results should be treated critically. The ribosome is assembled stoichiometrically, with a 1:1:1 proportion between the three types of rRNAs. There is no reports on excessive production of any type of rRNA, which inevitably had to be observed in case of such discrepancy. Moreover, the absolute numbers at the lowest extremes are obviously too small for the eukaryotic cell; 9 copies or 26 copies can be true values for a bacterium, but not for a mammal. The number of rRNA operons per bacterial genome varies from one up to 10+ depending on the species and strain growth speed (Klappenbach et al. 2000 DOI 10.1128/aem.66.4.1328-1333.2000), whereas the eukaryotic species have hundreds ribosomal repeat copies: in the unicellular Saccharomyces cerevisiae (yeast) the genes coding for the 35S rRNA precursor are organized in one cluster as about 100+ tandem repeats, and the metazoan have from 200 copies per genome in warm-blood animals up to several thousand in some species of plants, amphibian, and fish (Long, Dawid 1980 DOI 10.1146/annurev.bi.49.070180.003455). Thus, the numbers cited in the manuscript under reference #52 seems to be surely underscored.

Secondly, I would like to recommend supplementing the methods in '2. Detection of the rDNA variation' section. When determining the rDNA count per cell, it should be noted that PCR loses the game to hybridization-based techniques, such as FISH or non-radioactive dot hybridization, because replication (polymerization) is more sensitive to various template defects than hybridization. The GC-rich rDNA is opt to oxidation and forming non-canonical structures thus hindering replication process, while hybridization still occurs. Therefore, qPCR often underscores the rDNA count compared to dot-hybridization data. That is why the authors can not avoid to describe the classical method of silver staining, which was proved very effective to study NORs (AgNORs). Microscopy is briefly mentioned in the end of section 2 taking six lines (lines 114-119). However, FISH only is reviewed, with making no mention of cytogenetic techniques based on silver deposit assay.

The cytogenetic technique is a classical approach based on selective visualizing the NORs on metaphase chromosomes with silver nitrate (Ag-staining) (Goodpasture, Bloom 1975 DOI 10.1007/BF00329389; Howell, Black 1980 DOI 10.1007/BF01953855). The staining substrates are certain argentophilic proteins of the transcriptional complex (RNA polymerase I, UBF) that remain bound during metaphase only to those copies of ribosomal genes that have been transcribed in the preceding interphase. Thus, the process selectively visualizes clusters of transcriptionally active rDNA only (Hubbell 1985 DOI 10.3109/10520298509113926). The relative size of AgNOR in every given chromosome is constant in the cells of various tissues of the same individuals and is inherited as a codominant Mendelian character (Sozansky et al. 1984, 1985 DOI 10.1007/BF00293287, DOI 10.1007/BF00292588; de Capoa et al. 1991 DOI 10.1007/BF00206062; Velazquez et al. 1991 DOI 10.1139/g91-127). The AgNOR size is assessed visually in arbitrary units (rank assessment) (Lyapunova et al. 2017 DOI 10.1016/j.gene.2017.02.027). Using this approach, phenotypical effects of ribosomal gene abundance were studied in health and disease, as reviewed in (Porokhovnik, Lyapunova 2019 DOI 10.1007/s10577-018-9587-y).

Author Response

we thank you for your review report of our manuscript Variability of human rDNA and your important comments and provided references. We discussed them and revised as follows (the remarks requiring reply are in bold font):

Firstly, there is a questionable reference for the data related to the total copy number of ribosomal repeats in human genome. This is reference #52 with copy number data in lines 133-134: 'In one study the number of the ribosomal gene copies per cell varied in different individuals… from 9 to 421 for 5.8S… and from 26 to 282 for 28S.' Below, in lines 135-136, the authors expressed slight scepticism that the discrepancies could '…proceed from an error of measurement', but the data was generally presented as feasible. My opinion is that these results should be treated critically. The ribosome is assembled stoichiometrically, with a 1:1:1 proportion between the three types of rRNAs. There is no reports on excessive production of any type of rRNA, which inevitably had to be observed in case of such discrepancy. Moreover, the absolute numbers at the lowest extremes are obviously too small for the eukaryotic cell; 9 copies or 26 copies can be true values for a bacterium, but not for a mammal. The number of rRNA operons per bacterial genome varies from one up to 10+ depending on the species and strain growth speed (Klappenbach et al. 2000 DOI 10.1128/aem.66.4.1328-1333.2000), whereas the eukaryotic species have hundreds ribosomal repeat copies: in the unicellular Saccharomyces cerevisiae (yeast) the genes coding for the 35S rRNA precursor are organized in one cluster as about 100+ tandem repeats, and the metazoan have from 200 copies per genome in warm-blood animals up to several thousand in some species of plants, amphibian, and fish (Long, Dawid 1980 DOI 10.1146/annurev.bi.49.070180.003455). Thus, the numbers cited in the manuscript under reference #52 seems to be surely underscored.

We agree with the reviewer that the data in question are astonishing. In the revised version of the manuscript we tried to summarize his arguments in this manner: “But, if all these incomplete units are transcribed and processed, it is not clear how the stoichiometry of the ribosomal components is maintained. Also, it seems extraordinary that a person should have only 9 ribosomal genes per cell; this number would be normal for prokaryotes (Klappenbach et al. 2000, PMID: 10742207), but the homoiothermal animals usually have about 200 copies per genome (Long and Dawid 1980, PMID: 6996571).”

Still we would not say that the excess of a gene product should be “inevitably“ observed. The incomplete units may be silenced, or their transcripts may be quickly eliminated. On the other hand, if all RNAs from the abridged sequences are properly processed, the losses of individual genes may be mutually balanced in the common pool. Also, it should be noted that we do not know how many rDNA copies would be necessary to ensure normal metabolism of human cell.

Secondly, I would like to recommend supplementing the methods in '2. Detection of the rDNA variation' section. When determining the rDNA count per cell, it should be noted that PCR loses the game to hybridization-based techniques, such as FISH or non-radioactive dot hybridization, because replication (polymerization) is more sensitive to various template defects than hybridization. The GC-rich rDNA is opt to oxidation and forming non-canonical structures thus hindering replication process, while hybridization still occurs. Therefore, qPCR often underscores the rDNA count compared to dot-hybridization data. That is why the authors can not avoid to describe the classical method of silver staining, which was proved very effective to study NORs (AgNORs). Microscopy is briefly mentioned in the end of section 2 taking six lines (lines 114-119). However, FISH only is reviewed, with making no mention of cytogenetic techniques based on silver deposit assay.

The cytogenetic technique is a classical approach based on selective visualizing the NORs on metaphase chromosomes with silver nitrate (Ag-staining) (Goodpasture, Bloom 1975 DOI 10.1007/BF00329389; Howell, Black 1980 DOI 10.1007/BF01953855). The staining substrates are certain argentophilic proteins of the transcriptional complex (RNA polymerase I, UBF) that remain bound during metaphase only to those copies of ribosomal genes that have been transcribed in the preceding interphase. Thus, the process selectively visualizes clusters of transcriptionally active rDNA only (Hubbell 1985 DOI 10.3109/10520298509113926). The relative size of AgNOR in every given chromosome is constant in the cells of various tissues of the same individuals and is inherited as a codominant Mendelian character (Sozansky et al. 1984, 1985 DOI 10.1007/BF00293287, DOI 10.1007/BF00292588; de Capoa et al. 1991 DOI 10.1007/BF00206062; Velazquez et al. 1991 DOI 10.1139/g91-127). The AgNOR size is assessed visually in arbitrary units (rank assessment) (Lyapunova et al. 2017 DOI 10.1016/j.gene.2017.02.027). Using this approach, phenotypical effects of ribosomal gene abundance were studied in health and disease, as reviewed in (Porokhovnik, Lyapunova 2019 DOI 10.1007/s10577-018-9587-y).

We agree with the reviewer that silver staining of nucleolar organizer regions remains a useful method in the study of nucleoli. In our revised version, we mention it as follows: “Additionally, silver staining of the spread mitotic chromosomes may be used for the quantification of the transcriptionally active genes (Goodpasture and Bloom 1975, PMID: 53131; Howell and Black 1980, PMID: 6160049; Smirnov et al, 2006, PMID: 17089916; Porokhovnik and Lyapunova 2019, PMID: 30343462).” (The end of section 2).

But we cannot pursue this vein further without deviation from our topic. For, as the reviewer observed, only transcriptionally active genes are labelled by the silver. In our earlier publication (Smirnov et al, 2006; PMID: 17089916), combining FISH (rDNA + specific chromosomal probes) and silver staining, we found that e.g. two chromosomes #22 possessed gigantic NORs with a moderate silver staining; but much larger Ag-positive NORs were carried by four chromosomes #15, which showed a moderate FISH signal. In other words, one cannot rely on correlation between the number of copies and the number of active copies in the NORs. We also discovered that during the interphase the argyrophilic proteins in the FC/DFC units of the nucleoli undergo permanent fluctuation, which should be taken into account in the study of silver stained nucleoli (Hornacek et al, 2017, PMID: 28622108; Smirnov et al, 2020, PMID: 32119673). 

We hope that you find these revisions sufficient.

Reviewer 4 Report

This review by Smirnov et al deals with the variability of human rDNA. The authors first describe the methods allowing the detection of rDNA repeats variation. They then present the cause of rDNA variability, the distribution of variability in rDNA and the mechanism by which rDNA locus is stabilized despite extensive variability. They finally discuss the pathophysiological consequences of rDNA variation.

Overall, the topic covered by this review is  interesting. The authors discuss adequately the current data. As an improvement, although I appreciate that the review is focused on human rDNA variability, I think that in some instance useful information could be obtained through comparison with model organisms.

Other points:

The authors could consider presenting the fourth paragraph immediately after the paragraph on the detection of rDNA variations. As it stands, I think that it cut the flow between the cause of rDNA variability and the mechanisms allowing its stabilization.

Page 4: the sentence “it has been suggested that if variation in copy number is induced by transcription, then the latter may be responsible for the increased mutability of the rapidly proliferating cells” is not clear to me. Please rephrase.

Page 6, end of first paragraph. The authors should precise what the authors have found in the sentence:

“But a recent study suggested that they may be involved in regulation of the copy number in the NOR”. Indeed, as it stands, they seem to doubt about the findings.

Author Response

we thank you for your review report of our manuscript Variability of human rDNA and your important comments and provided references. We discussed them and revised as follows (the remarks requiring reply are in bold font):

As an improvement, although I appreciate that the review is focused on human rDNA variability, I think that in some instance useful information could be obtained through comparison with model organisms.

In the revised version of the manuscript we added references concerning prokaryotes and various metazoan organisms (Klappenbach et al, 2000, PMID: 10742207; Long and Dawid, 1980, PMID: 6996571; Salvi et al, 2020, PMID: 25073155) 

The authors could consider presenting the fourth paragraph immediately after the paragraph on the detection of rDNA variations. As it stands, I think that it cut the flow between the cause of rDNA variability and the mechanisms allowing its stabilization.

In arranging our material, we could not separate causes of the variability from their kinds. But the latter are closely linked to the distribution (for instance, recombination and its hot spots), whereas the stabilization of the rDNA locus seems to us a more independent subject. Therefore, we prefer the original disposition of the sections.

Page 4: the sentence “it has been suggested that if variation in copy number is induced by transcription, then the latter may be responsible for the increased mutability of the rapidly proliferating cells” is not clear to me. Please rephrase.

We rephrase it as follows: “It has been suggested that the dosage variation induced by the intensive transcription may be responsible for the increased mutability of the rapidly proliferating cancer cells.“

Page 6, end of first paragraph. The authors should precise what the authors have found in the sentence: “But a recent study suggested that they may be involved in regulation of the copy number in the NOR”. Indeed, as it stands, they seem to doubt about the findings.

In the revised version, we add (after the quoted phrase) the following passage: „‘Accordingly, some authors propose to distinguish between “scheduled regulatory” and “unscheduled harmful” R-loops (Niehrs and Luke, 2020, PMID: 32005969). Data of a recent study indicate that antisense transcription by nucleolar RNA polymerase II generates an “R-loop shield” at IGS, which protects nucleolar organization by suppression of disruptive ncRNAs transcribed by pol I (Abraham et al, 2020, PMID: 32669707).‘

We hope that you find these revisions sufficient.